# Computational Strategies for Scalable Genomics Analysis

**DOI:** 10.3390/genes10121017

**Published:** 2019-12-06

**Authors:** Lizhen Shi, Zhong Wang

**Affiliations:** 1Department of Computer Science, Florida State University, Tallahassee, FL 32304, USA; lizhen9.shi@gmail.com; 2US Department of Energy, Joint Genome Institute, Walnut Creek, CA 94598, USA; 3Environmental Genomics and Systems Biology Division, Lawrence Berkeley National Laboratory, Berkeley, CA 94720, USA; 4School of Natural Sciences, University of California at Merced, Merced, CA 95343, USA

**Keywords:** scalable genomics analysis, big data, high performance computing, cloud computing

## Abstract

The revolution in next-generation DNA sequencing technologies is leading to explosive data growth in genomics, posing a significant challenge to the computing infrastructure and software algorithms for genomics analysis. Various big data technologies have been explored to scale up/out current bioinformatics solutions to mine the big genomics data. In this review, we survey some of these exciting developments in the applications of parallel distributed computing and special hardware to genomics. We comment on the pros and cons of each strategy in the context of ease of development, robustness, scalability, and efficiency. Although this review is written for an audience from the genomics and bioinformatics fields, it may also be informative for the audience of computer science with interests in genomics applications.

## 1. Introduction

The successful completion of the human genome project led to a revolution in DNA sequencing. In the ten years following the completion of the first human genome in 2003, several sequencing technologies, collectively called Next-Generation Sequencing (NGS), underwent several rounds of development and displacement. Each round brought capacity increases of 100–1000 fold (reviewed in [1]). By 2019, two types of sequencing technology remain popular: Short-read and long-read. If we think of a genome as a book, short-read sequencers read this book sentence-by-sentence with few errors, while long-read sequencers read it paragraph-by-paragraph, typically with a higher error rate. Illumina dominates the short-read sequencing market and two companies presently compete in the long read sequencer market, Pacific Biosciences and Oxford Nanopore. All platforms are massively parallel sequencers. Currently, the majority of sequencing data are from short-read technologies because they have much higher throughput and lower cost per base. Thanks to these powerful sequencing technologies, genomics, like many other disciplines, has entered the big data era [2]. In FY2018, the Department of Energy Joint Genome Institute (DOE JGI) generated 200 Terabases (Tb) of sequence data, enough to sequence one human genome over 3000 times at 20× coverage. The total amount of sequence of the first release of the UK BioBank project (50,000 participants, or about 1/10 of the total) is already over 50 TB [3].

Unlike other domains, where big data are characterized by the four “V”s: Volume (for large scale), variety (for different forms), velocity (for fast streaming) and veracity (for data uncertainty), genomics big data also has a “U” characteristic, meaning the majority of genomics data are unstructured. In addition, data volume, variety, and veracity often increase as genomics data flows through various data analytic steps. It is not uncommon to need disk space 100–200 times the input for storing temporary data. Software tools, having very diverse input requirements and functionalities, are needed to extract and reduce the information contained in the sequence data, and these tools inevitably introduce noises and biases.

The exponential growth of genomics data poses a great challenge for data management and analysis. We will not discuss data storage and management problems here. Interested readers are encouraged to consult the excellent review [4]. In this review, we will discuss general strategies of scalable genomics solutions without focusing on algorithms for specific genomics applications. We will discuss these solutions in the context of four criteria: Ease of development, robustness, scalability, and efficiency. Here, scalability refers to the capability of a bioinformatic tool to handle bigger data/problems when additional resources are added. Efficiency indicates how well a tool utilizes the computational resources of the system. We use scalability in combination with efficiency to measure how well a bioinformatic tool effectively uses increasing numbers of parallel processing elements (nodes, CPUs, cores, processes, threads, etc.) for processing a growing amount of data. It is important to note that, besides parallelization that we are going to discuss below, there are many generic factors that could impact the performance of genomic solutions, including compilers, programming models, software-hardware optimization, etc.

## 2. Scalable Strategies

### 2.1. Shared-Memory Multicore Architecture

Given that many genomics research groups do not have seasoned software engineers for distributed software development, exploiting single servers with a large amount of RAM and CPU cores is a straightforward solution for large-scale sequence analysis. My group initially invested in big memory nodes with Terabytes of RAM. The cache-coherent non-uniform memory access architecture of the XSEDE resource Blacklight, housed at the Pittsburgh Supercomputing Center (PSC), with up to 16 TB of RAM, enables very large plant genome assemblies such as the hexaploid wheat genome [5]. Similarly, an SGI UV200 machine with 7 TB of shared RAM also enables wheat genome assembly (38 days on 64 CPUs) [6]. Amazon Web Services (AWS) also offers large memory EC2 instances with up to 4 TB and 128 cores (X1e instances) and much larger memory footprints are in alpha testing.

In these shared memory supercomputers with large memory and lots of cores, parallelism is usually achieved by multi-threading, i.e., dividing a process into multiple threads and executing them in parallel. As memory and other resources are shared among threads, threads can read and write to the shared memory without the need for special mechanisms of communication. Pthread [7] and OpenMP [8] are two implementations of multi-threading at different levels. Pthreads, the POSIX standard thread library, is a low-level application programming interface (API) for working with threads, whereas OpenMP gives programmers higher level threading options that greatly simplify programming and debugging. Consequently, OpenMP is the predominant threading model on shared memory systems for scientific computing. Accordingly, there are more OpenMP-based tools (SPAdes [9,10,11]) than Pthread-based tools (ALLPATHS-LG [12], SOAPdenovo [13] ) on genomic analysis.

These large memory systems enable rapid results without extra software development time. However, the cost of upgrading a node increases exponentially with memory size. Furthermore, there are physical limits to how much memory or cores can be added to a node.

### 2.2. Special Hardware

The increasing demand for power-efficient, high-performance computing spurred a revolution in computer architectures over the last couple of decades, as specialized processing units emerged to improve the efficiency of parallel tasks. In this environment, the classical host processor delegates the execution of the computationally intensive parts of the job to co-processors, such as field-programmable gate arrays (FPGA) [14], graphics-processing units (GPU) [15], and tensor processing unit (TPU) [16], to speed up the overall execution. These special hardware architectures are now being applied to genomics data analysis with remarkable success. For example, Falcon Computing developed an FPGA-based solution [17] that speeds up the genome analysis tool kit (GATK) [18], a genomics variant analysis suite, by 50 times. GPU has a long history in computational biology for special applications such as molecular dynamics simulation [19,20], and recently sees an increasing application in NGS data analysis (reviewed in [21]). As the genomics field is rapidly adopting deep learning technologies (reviewed in [22]), GPUs and TPUs are expected to speed up more genomics applications such as AlphaFold [23].

These special hardware architectures greatly increased the amount of parallelism that a machine can exploit. However, there are several limitations including availability, difficulty scaling on heterogeneous systems, and the need to port existing CPU-based algorithms to these systems. In addition, training large deep neural networks on GPUs/TPCs could be very cost-prohibitive [24].

### 2.3. Multi-Node HPC Scalability

For on-premise hardware scaling, one could either scale up by upgrading the existing nodes with more capacity (cores, memory, co-processing unit, storage, etc.) as we discussed above, or scale out—by adding more nodes to form a high-performance computing cluster (HPC).

*Message Passing Interface* (MPI) [25], the *de facto* industry standard for distributed memory systems, is a language-independent communications protocol that uses a message-passing paradigm to share the data and state among a set of cooperative processes running on an HPC cluster. As it takes the advantage of data locality, MPI-based implementation yields great computing performance compared with alternatives [26]. MPI-based NGS sequence analysis tools, including read aligner such as pBWA [27] and assembler such as Ray [28], can scale up to hundreds of thousands cores on a HPC cluster.

*Partitioned Global Address Space* (PGAS) [29] is a distributed shared-memory programming model that combines the advantages of the shared-memory programming paradigm and the performance of the message passing programming paradigm. Unified Parallel C (UPC) [30] and UPC++ [31] are C and C++ extensions of PGAS model, repsectively, by combining the advantages of the PGAS and C/C++ language features such as templates, object-oriented design, operator overloading, and lambda functions. These advantages have brought noticeable performance gains in a few challenging genomic problems including metagenome assembly: UPC-based tools like Meta-HipMer [32] can assemble a 2.6 TB metagenome dataset in just 3.5 h with 512 nodes.

The biggest drawback of MPI and the PGAS languages is their programmability, as they require experienced software engineers to take care of fine-grained control over mechanisms such as memory locality, data communication, and tasks synchronization. These inevitably drive up the development and maintenance costs. Another potential drawback of large-scale runs is fault-tolerance; failure of one process could lead to the failure of the entire application. Some recent community efforts aim to combine the ease of programming such as Python with the superior efficiency of distributed computing paradigms such as MPI, e.g., the RAY project (https://github.com/ray-project/ray), which may encourage more genomics applications to take advantage of the distributed systems.

### 2.4. Cloud Scalability

The need to process an unprecedented amount of genomics data efficiently and robustly also drives many applications built on cloud computing technologies (reviewed in [33]). In cloud computing paradigms, data are distributed to a large number of nodes and computation is shifted to the node where the data resides. Hadoop and Spark are two powerful big-data frameworks for cloud computing.

The Hadoop framework [34] is the Apache’s open-source implementation of Google’s MapReduce [35] programming model. With the combination of its two core components, Hadoop Distributed File System (HDFS) [36] and MapReduce, Hadoop enables a load-balanced, scalable, and robust solution for big data analytics. Many Hadoop-based applications have been developed for genomics, to name a few, NGS read alignment [37,38], genetic variant calling [39], sequence analysis [40,41].

The IO-intensive nature of Hadoop’s MapReduce can severely limit its performance. For example, map tasks of genomics applications often produce 10–100× amount of intermediate data stored in local disks until the reduce tasks remotely fetch them (pull) via HTTP connections, adding very significant communication overhead. Since there is no state shared between individual map and reduce tasks, Hadoop is not suitable for iterative tasks. Reusing the same dataset across multiple iterations is very common in sequence analysis algorithms, making Hadoop-based solutions very inefficient compared to implementations of MPI and OpenMP.

Apache Spark [42] was developed to overcome the above limitations of Hadoop by providing an efficient abstraction for in-memory computing called Resilient Distributed Dataset (RDD) [43]. Spark can hold the intermediate data and computations in memory as persisted RDDs, thereby improving performance by reducing disk overhead. Besides Scala, Spark also supports other programming languages such as Python and R, which greatly improves its programmability. Integration with the Jupyter notebook interactive environment, or a managed platform provided by Databricks [44], can significantly shorten software development and data analysis time. Many Spark-based genomics applications have been developed for large-scale sequence processing on public or private cloud systems [45,46,47]; for a comprehensive review, please refer to [45].

Spark with in-memory data processing is significantly faster than Hadoop, but it is still slower than MPI-based implementation, largely owing to the latter’s low-level programming language and reduced overhead (e.g., no fault handing like Spark has). Another challenge is that not all the components in a complex genomics data processing pipeline can be easily ported to Spark due to the lack of corresponding libraries.

Cloud-based genomics solutions are evolving in a rapid pace. Cloud systems integrate data management (store, access, and share) and data analysis into one platform, providing flexibility to scale in/out and up/down, and offer user-friendly, consistent, reproducible data pipelines. Such solutions, e.g., Terra (https://app.terra.bio/), will soon unshackle genomics data scientists from the burden of managing hardware and software infrastructures and enable large team collaborations.

### 2.5. Container Scalability

Bioinformatic pipelines often consist of several independent tools or modules, and these components may require different running environments or have different software dependencies, making them difficult to deploy, manage, and run. Containerization packages all components of a pipeline and their dependencies into a container image so that the pipeline can run consistently on any infrastructure across on-premise and cloud environments. Containers are gaining much popularity because it addresses the challenges of sharing bioinformatics tools and enabling reproducible analyses. For example, ORCA (the genomics Research Container Architecture [48]) provides containers of over 600 bioinformatics tools (as of October 2019). Another resource, Dockerhub (hub.docker.com), also hosts 100+ genomics-related containers. Among the available container platforms, Docker [49] is by far the most popular choice. One major limitation of Docker is its root-owned daemon process, which is not acceptable on most HPC systems. Shifter [50] and Singularity [51] are docker-alternatives that support HPC-based containers.

Containers are “units of deployment”. One can host more containers on a single node by upgrading its hardware resources (scale-up). For heavy workloads, Kubernetes [52] (often abbreviated just as “k8s”) is a container-orchestration system that scales out container services on a cluster of nodes. It achieves this by abstracting infrastructure components coordination, auto-scaling, and self-healing. Kubernetes works with many container engines and enables automated deployment on cloud or on-premise clusters. One of the successful adoptions of Kubernetes in genomics is the Galaxy project [53]. However, Kubernetes does not reduce the deployment complexity for applications with complex workflows. To overcome this limit, one solution is to encapsulate a set of Kubernetes resources into a preconfigured package, or charts, and manage the charts using a manager like Helm [54]. Another solution is to use a container-native workflow engine such as Argo [55], which supports both directed acyclic graph (DAG) and step-based workflows on Kubernetes. For deploying machine-learning workflows, Kubeflow [56] makes it simple, portable, and scalable. Despite these efforts, compared to well-established resource managers and schedulers such as slurm or torque in HPC and YARN in the cloud, Kubernetes is still in its infancy.

## 3. Conclusions and Future Perspectives

We were only able to cover a few scaling strategies for genomics analysis in Section 2, summarized in Table 1 in the context of ease of development, robustness, scalability, and efficiency. In the table, we also listed some bioinformatic tools using these strategies. One might have to combine several strategies, for example, using a single node for development and HPC for large-scale production. When the scale of the analysis exceeds one’s on-premise capacity, they can "spillover" to cloud-based solutions for additional capacity.

It is important to note that genomic analysis pipelines are complex sequential-parallel systems consisting of multiple bioinformatic tools, each with a variable degree of parallelism. The overall scalability is often limited by the step with the least scalability. In addition, the performance gain from increasing parallelism is limited by the Amdahl’s law [65] in both parallel and distributed systems.

Serverless, or “Function-as-a-Service”, is another interesting concept that is currently undergoing rapid development. It eliminates the need to set up computing infrastructure and brings several benefits including ease-of-use, instantaneous scalability, and cost-effectiveness. Breaking up a gigantic monolithic application into smaller micro-services enables scaling up individual functions, reducing development time, and increasing agility as the pipeline evolves faster. We expect more and more developers to deploy their complex bioinformatics workflows as serverless web services by taking advantage of the rich cloud ecosystems offered by major commercial cloud vendors (AWS Lambda, Microsoft Azure, and Google Clouds Function, etc.)

## Figures and Tables

**Table 1 genes-10-01017-t001:** A Comparison of Current Scalable Technologies.

	Shared-Memory Multicore Architecture	Special Hardware	HPC	Cloud	Container Orchestration
Ease of dev	+++	+	+	++	+++
Robustness	++	++	+	+++	+++
Scalability	+	+	+++	+++	+++
Efficiency	++	+++	++	+	+
Representative Software	SPAdes [9], OpenMP LCS [11], ALLPATHS-LG [12], SOAPdenovo [13], LSHvec [57]	FAGP [17], SOAP3 [58], SOAP3-dp [59], CUSHAW2-GPU [60] FPGA-based Smith-Waterman [61]	pBWA [27], Ray [28], Meta-HipMer [32] ClustalW-MPI [62] TREE-PUZZLE [63]	HAlign [37] BigBWA [38], BioPig [40,41], SpaRC [46], Metaspark [47], SparkBWA [64]	Galaxy-Kubernetes [53]

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
