# Peer review of "Computational Strategies for Scalable Genomics Analysis"

_genes, 2019, doi:10.3390/genes10121017_

Round 1
Reviewer 1 Report
The review covers many parallel programming methods including shared and distributed memory approaches but also technology that supports these.
I would suggest the authors to clarify better “Scalability” since it both implies speedup and scaling problem size. For instance, shared memory approaches provide more speedup potential but limited to problem size (unless hardware changes, e.g., more memory). Distributed memory approaches allow to scale the problem (by splitting work between workers) with the cost of collective communication (which depends on the algorithm).
The review also could mention Amdahl’s law[1], as this would give readers a better understanding how much performance should be expected from the fraction of the algorithm that can be ran parallel. How scalable are Genomic Analysis methods?
Finally, speedup is not only exclusive to parallelization, compilers, programming models, software-hardware optimization are all related to HPC, with a cost on programmability.
[1] Hill, M. D., & Marty, M. R. (2008). Amdahl’s Law in the Multicore Era. Computer, 41(7), 33–38. https://doi.org/10.1109/MC.2008.209
Author Response
Q: I would suggest the authors to clarify better “Scalability” since it both implies speedup and scaling problem size. For instance, shared memory approaches provide more speedup potential but limited to problem size (unless hardware changes, e.g., more memory). Distributed memory approaches allow to scale the problem (by splitting work between workers) with the cost of collective communication (which depends on the algorithm).
A: Thanks for pointing this out. We revised our manuscript accordingly. Please refer to the last paragraph in the introduction.
Q: The review also could mention Amdahl’s law[1], as this would give readers a better understanding how much performance should be expected from the fraction of the algorithm that can be ran parallel. How scalable are Genomic Analysis methods?
A: This is a good point. We revised our manuscript accordingly. Please refer to the first paragraph in the conclusion and future perspectives.
Q: Finally, speedup is not only exclusive to parallelization, compilers, programming models, software-hardware optimization are all related to HPC, with a cost on programmability.
A: We revised our manuscript accordingly to make this point clear. Please refer to the last paragraph in the introduction.
Reviewer 2 Report
Overall the manuscript is well written. The authors provide a concise overview of the computational strategies relating to the scalable analysis of NGS data. I have only one comment:
Given that the manuscript deals with scalability of NGS data, I think the manuscript is missing a paragraph discussing the use of kubernates/singularity/docker for open source container orchestration in cloud based compute.
Author Response
Q: Given that the manuscript deals with scalability of NGS data, I think the manuscript is missing a paragraph discussing the use of kubernates/singularity/docker for open source container orchestration in cloud based compute.
A: Thanks for pointing this out. We added a subsection named container scalability in section 2 accordingly.